



# Effects of storage temperature on physiological characteristics and vegetative propagation of desiccation-tolerant mosses

Yuewei Guo and Yunge Zhao

State Key Laboratory of Soil Erosion and Dry-land Farming on the Loess Plateau, Institute of Soil and Water Conservation, Northwest A & F University, Yangling, 712100, Shaanxi, China

*Correspondence to*: Yunge Zhao (zyunge@ms.iswc.ac.cn)

**Abstract.** Mosses, as major components of later successional biological soil crusts (biocrusts), play many critical roles in arid and semi-arid ecosystems. Recently, some species of desiccation-tolerant

mosses have been artificially cultured to speed up the recovery of biocrusts. Revealing the influencing factors on the vegetative propagation of mosses will benefit the restoration of moss crusts, which is an important reproductive mode of mosses in arid and semi-arid region. In this study, three desiccation-tolerant mosses (*Barbula unguiculata, Didymodon vinealis* and *Didymodon tectorum*) were stored at five temperature gradients (0 ℃, 4 ℃, 17 ℃, 25 ℃ and 30 ℃) for 40 days. Then vegetative

propagation and physiological characteristics of the three mosses were investigated to determine the influence of storage temperature on vegetative propagation of desiccation-tolerant mosses and its mechanism. The results showed that vegetative propagation of the three mosses varied with temperature, and the most significant change was observed in *D. tectorum* after storage at different temperatures. Conversely, no significant difference was found in *D. vinealis*. Only germination

percentage of *B. unguiculata* was not significantly different at all storage temperatures. The enhancement in regenerative capacity of the three mosses was accompanied by an increased temperature from 0 ℃ to 17 ℃ and a decrease beyond that. Malondialdehyde (MDA) content of the three mosses was increased by more than 50% at all of the investigated temperatures, meanwhile, soluble sugar content increased in the three mosses. However, a decrease trend was observed in MDA

content from 0 to 17 ℃. As the temperature increased, the contents of chlorophyll and soluble protein in *B. unguiculata* increased, while decreased in *D. vinealis* and *D. tectorum*. The integrity of cell and its membrane probably is the important influencing factor on the vegetative propagation of desiccation-tolerant mosses. Although a 40-day storage period caused cell injury, our results suggested that storage temperature could enhance or suppress such injury and change vegetative propagation

capacity of the three mosses. It could be concluded that the suitable storage temperature of *B. unguiculata* was 4 ℃ and the optimal temperature was 17 ℃ for *D. vinealis* and *D. tectorum*.

## 1 Introduction

Biological soil crusts (biocrusts) are composed of microscopic (cyanobacteria, algae, fungi, and bacteria) and macroscopic (lichens, mosses) poikilohydric organisms that occur on or within the top

few centimeters of the soil surface (Belnap et al., 2016). They are widely distributed and play important roles in many arid and semi-arid ecosystems, such as soil surface stabilization, soil fertility





accumulation and soil hydrology regulating (Belnap and Lange, 2003). As major components of later successional biocrusts, mosses exerted much stronger ecological functions than cyanobacteria (Seppelt et al., 2016; Gao et al., 2017; Lan et al., 2012). Thus, some researchers suggest culturing artificially moss biocrusts on the degraded soil surface so as to speed up the recovery of degraded arid and

semi-arid ecosystems (Belnap and Eldridge, 2003; Zhao et al., 2016). Recently, some mosses have been investigated by using the theory of vegetative reproduction (Jones and Rosentreter, 2006; Xiao et al., 2015). However, little is known about the vegetative propagation of such mosses. It may be an important reason why the cultivation research of moss crusts is still tentative.

Vegetative propagation is an important reproduction mode of bryophytes (hornworts, liverworts

and mosses) in dry habitats, and gametophyte fragments may serve as the dominant inoculum in mosses (Mishler, 1988; Tian et al., 2005). So far, a number of moss cultivation experiments that used gametophyte fragments have been conducted to establish new colonies in laboratory and field (Cleavitt, 2002; Jones and Rosentreter, 2006; Xiao et al., 2015). All of such studies demonstrated that artificial cultivation could speed up the succession process of moss crusts. For example, Antoninka et al. (2016)

found the coverage and biomass of mosses on the artificially inoculated soil surface increased faster than that of uninoculated. Furthermore, some researchers suggested that inoculation material should be mass-produced by vegetative regeneration with rapid development (Jones and Rosentreter, 2006; Mishler, 1988) because of the necessity when moss biocrusts would inoculate in large areas. Although influencing factors of tissue cultivation of mosses were investigated for a long time (Duckett et al.,

2004; Hoffman, 1966; Sabovljevic et al., 2003), the mechanism of mosses regeneration still needs to be further studied.

Desiccation-tolerance (DT) is a remarkable feature of biocrust mosses. The feature makes mosses can lose virtually all of the free intra-cellular water and recover normal function after rehydration (Proctor et al., 2007). Because metabolism of desiccation-tolerant mosses is suspended under water

stress and the cell integrity can be maintained during the dry periods (Mansour and Hallet, 1981; Platt et al., 1994). Then cellular activity of mosses will resume and return to the normal hydrated state within a few minutes to a few hours after being rehydrated (Platt et al., 1994; Pressel et al., 2006). Therefore, mosses could regenerate even after more than a decade of dry storage (Bristol, 1916; Keever, 1957). However, decline and disappearance in the regenerative capacity of *Grimmia laevigata* exerted that

mosses might suffer irreversible damage in a long-term desiccation (Keever, 1957). It was still unclear why the potential of vegetative propagation of mosses was altered by storage, or the recovery abilities between moss species were different after drought dormancy, which would impede the study of moss cultivation.

In general, DT investigations concentrate more on its mechanism and evolutionary history

(Proctor et al., 2007; Oliver et al., 2000), and less on artificial cultivation. Actually, there are lots of theoretical studies that can support further utilization. For instance, the impact of desiccation stress on moss regeneration varies with drying time and storage temperature (Keever, 1957; Burch, 2003) may guide research on the regenerative mechanism of mosses upon desiccation and artificial cultivation of mosses. Furthermore, DT plays essential roles in mosses regeneration in dry habitats, which reminder

us to investigate the link between physiological characteristics and vegetative propagation of the





mosses. Based on the study results mentioned above, it can be hypothesized that (1) dry storage may impact vegetative propagation of desiccation-tolerant mosses, (2) the change of vegetative propagation after storage may relate to the influence of storage on physiological characteristics of mosses, and (3) the effect degree of storage on vegetative propagation and physiological characteristics is related to the storage temperature.

Consequently, in this study, three desiccation-tolerant mosses including *Barbula unguiculata*, *Didymodon vinealis* and *Didymodon tectorum*, which were the dominated mosses in biocrusts communities in the Loess Plateau region, were selected and stored at five temperature gradients (0 ℃, 4 ℃, 17 ℃, 25 ℃ and 30 ℃) for 40 days. Then, (1) the effect of storage temperatures on vegetative propagation of the three mosses and (2) the change of physiological indexes, including contents of chlorophyll, soluble sugar, soluble protein and malondialdehyde (MDA) were investigated so as to reveal the influence of storage temperature on the vegetative propagation of mosses and its mechanism.

## 2 Materials and methods

### 2.1 Moss species and their collection

Moss taxa used in the study were *Barbula unguiculata*, *Didymodon vinealis* and *Didymodon tectorum*. The mosses were collected from moss-dominated biocrusts with coverage around 80% on north-facing slopes in the study region. The study was conducted in Ansai Country, Shaanxi Provence, China (109°19' E, 36°51' N), which located in the central part of the Loess Plateau. Elevation of the sampling plot varies from 1,068 to 1,309 m. The plot has a typical semi-arid continental climate with the average annual temperature of 8.8 ℃ and the range of monthly average temperature is from 22 ℃ to -7 ℃ in July and January, respectively. The average annual accumulated temperature above 0 and 10 ℃ are 3733 and 3283 ℃, respectively. The average annual precipitation is 500 mm with 60% or more falling between June and September (Zhang et al., 2011). In fact, the average monthly precipitation was 11.98 mm when the moss crusts were collected in November 2016, and the average monthly temperature ranged from 9.88 to −3.64 °C (Chinese Central Meteorological Station, 2017). The moss crusts were air dried (all of the water content of mosses less than 10%) in the shade as soon as they were collected. Then, they were transported to the laboratory of State Key Laboratory of Soil Erosion and Dry-land Farming on the Loess Plateau which is in Yangling, Shaanxi Province.

### 2.2 Experimental design

Each of the three moss crusts was separated into two parts as soon as they were transported to the laboratory. One was used to measure initial physiological indexes (chlorophyll content, soluble sugar content, soluble protein content and MDA content) and germination parameters (gametophyte germination, gametophyte increment and gametophyte vigour index). The other was stored at five temperature gradients, i.e., 0 ℃, 4 ℃, 17 ℃, 25 ℃ and 30 ℃. All the temperatures were controlled within ± 1 ℃ around the target. Three separated subsamples (duplicates) of each moss species were stored at each temperature gradient. Before being storing, the moss samples were packed in ziplock baggies to block the change of water content, and then they were kept in the dark under light-blocking





fabric. Then, the mosses were taken out on the 41st day of storage, and the physiological indexes and germination parameters as mentioned above were measured.

### 2.3 Measurement of physiological index and germination parameters

### 2.3.1 Physiological index

5    The living mature gametophytes of *B. unguiculata*, *D. vinealis* and *D. tectorum* were collected from moss crusts for measuring contents of chlorophyll, soluble sugar, soluble protein and MDA shortly after rehydrated and washed with deionized water. Approximately 0.1 g fresh mass of gametophytes was used to measure the contents of soluble sugar, soluble protein and MDA; while approximately 0.05 g fresh mass was used for measurement of chlorophyll content. The four indicators were measured by 10   the following protocols with three replications.

    The chlorophylls were extracted by 95% (v/v) ethanol and then boiled the solution at 85 ℃ for 5 min. After being centrifugation at 4000 rpm for 10 min, chlorophylls in the supernatant were measured the absorbance at 665 and 649 nm with the spectrophotometer (UV-2300, *Techcomp*, China) (Wellburn and Lichtenthaler, 1984).

15     After the soluble protein was extracted into ice-cold 50 mmol $L^{-1}$ phosphate buffer (pH 7.8), the supernatant was collected after being centrifugation at 8000 rpm for 30 min at 4 ℃. The soluble protein stained with Coomassie brilliant blue G-250 and read absorbance at 595 nm (Bradford, 1976).

    The MDA as well as soluble protein was extracted and centrifuged. Then the supernatant was homogenized with 0.6% (W/V) thiobarbituric acid dissolved by 1 mol $L^{-1}$ NaOH and 10% (W/V) 20   trichloroacetic acid. The mixed solution was heated at 100 ℃ for 20 min, and then read absorbance at 450, 523 and 600 nm (Hodges et al., 1999). The *Techcomp* UV-2300 spectrophotometer was also used to measure the absorbance of MDA and soluble protein.

    The soluble sugar was extracted by distilled water at 100 ℃ for 30 min. After being filtered and diluted, the extract was added to anthrone–sulfuric acid solution. The mixed solution was used to 25   measure absorbance at 620 nm with the spectrophotometer (UV-1601, *Shimadzu*, Japan) (Morris, 1948).

    The fresh weight of gametophytes was measured shortly after rehydration, and then their dry weight was measured after oven drying to constant weight at 70 ℃ (Schonfeld et al., 1988). Both of them were used to calculate the four physiological indexes on dry basis.

30   **2.3.2 Germination parameters**

    At the same time of measuring physiological indexes, some gametophytes of the three moss species were collected to test the germination parameters. The loessial soil (uniform soil texture of *Calciustepts*) collected from the study region was used for culturing the mosses. The soil was sieved through a 0.25-mm mesh and placed in each pore of a 6-well plate, whose diameter is 35 mm and depth is 12 mm. 35   Then soil was adjusted water content by deionized water to 23% (W/W) (the field water holding capacity of the soil). The top 2 mm of living mature gametophytes of the mosses were cut, rehydrated, washed and inoculated on the flatted surface of the soil. Five inocula were placed separately in one well. Thus, 30 inocula were inoculated in each 6-well plate as one replication. Three 6-well plates were set





for each moss species. Totally, 90 experimental inoculations were set for the measurement of germination parameters before and after being stored at the five temperature gradients for each moss species. Meanwhile, three 6-well plates without inoculated mosses were set as controls in the experiment in order to eliminate the effect of other propagules, like spores in the soil used. The 6-well

plates were wrapped tightly with transparent plastic films for holding soil moisture. After that, they were put into a growth chamber (AGC-D003N, China) to incubate. Parameters of the growth chamber were set as 12-h photoperiod (4500-5500 Lux), constant temperature of 17 ℃ (± 1 ℃) and relative humidity of 60-70%. During the period of incubation, deionized water was supplied so as to keep soil moisture at 23%. The new gametophytes were counted every five days since the first day when they

were found. There were five observations altogether during the next 25 days. It was noteworthy that no new gametophyte was found in the blank 6-well plates during all of incubation in the study.

By analogy with seeds germination, the vegetative propagation of moss gametophytes was described by three germination parameters, including gametophyte germination, gametophyte increment and gametophyte vigour index. In this paper, the gametophyte germination means the

percent of moss inocula germinated. The gametophyte increment means the average of new gametophytes in a 6-well plate. The gametophyte vigour index refers to seed vigour index (Abdul-baki and Anderson, 1973). The germination percentage of seed and length of hypocotyl were replaced by the gametophyte germination and average of new gametophytes, respectively, in the gametophyte vigour index. Then germination parameters were calculated by Eqs. (1) - (3):

$$\text{gametophyte germination} = \frac{\text{number of germinated inocula}}{\text{number of total inocula}} \times 100\% \qquad (1)$$

$$\text{gametophyte increment} = \frac{\text{number of new gametophyte}}{\text{number of total inocula}} \qquad (2)$$

$$\text{gametophyte vigour index} = \text{gametophyte germination} \times \text{gametophyte increment} \qquad (3)$$

According to Eqs. (1)-(3), the gametophyte vigour index could describe summarily the vegetative propagation of the mosses.

**2.4 Statistical analyses**

The differences of physiological indexes and germination parameters were tested using one-way analysis of variance (ANOVA) with Fisher's least significant difference post hoc test (LSD) at $P < 0.05$. The relationships between physiological indexes and germination parameters of the three moss species were quantified by calculating Pearson correlation coefficient. These statistical analyses were

completed using SPSS 22.0.

The effect of physiological characteristics on vegetative propagation was analyzed by grey incidence analysis in Microsoft Excel 2010 (Deng, 1984; Lin et al., 2009). The grey incidence degree between the reference sequences (physiological indexes) and the compared sequence (gametophyte vigour index) were calculated by Eqs. (4) - (6):

$$\Delta_i(k) = |y(k) - x_i(k)|, k = 1, 2, \ldots \ldots, \text{n}; i = 1,2,3,4 \qquad (4)$$

$$\xi_i(X_i, \ Y) = \frac{\min_i \min_k \Delta_i(k) + \rho \max_i \max_k \Delta_i(k)}{\Delta_i(k) + \rho \max_i \max_k \Delta_i(k)}, k = 1, 2, \ldots \ldots, \text{n}; i = 1,2,3,4 \qquad (5)$$

$$r_i = \frac{1}{n} \sum_{k=1}^{n} \xi_i(k), k = 1, 2, \ldots \ldots, \text{n}; \ i = 1,2,3,4 \qquad (6)$$



where $\Delta_i(k)$ and $\xi_i(X_i, Y)$ are the absolute difference and the grey relational coefficient, respectively, between $X_i$ (physiological indexes) and $Y$ (gametophyte vigour index) at point $k$. The grey relational coefficient ($r_i$) is between the $i_{th}$ physiological index and its gametophyte vigour index when the distinguishing coefficient ($\rho$) is 0.5.

The grey incidence degree is a sum of the grey relational coefficients.

## 3 Results

### 3.1 The initial state of the mosses

The initial physiological indexes and germination parameters of the three mosses were shown in Table 1. It could be seen that the four physiological indexes and gametophyte germination of *D. vinealis* were

significantly higher than the other two species. The biggest gametophyte increment and gametophyte vigour index were found in *D. tectorum* and the smallest germination parameters were found in *B. unguiculata*. However, no significant difference in the contents of chlorophyll, soluble protein and MDA between *D. tectorum* and *B. unguiculata* was found.

**Table 1** The initial value of physiological index and germination parameters in the three mosses

| index | *B. unguiculata* | *D. vinealis* | *D. tectorum* |
|---|---|---|---|
| chlorophyll content (mg g$^{-1}$) | 1.53 ±0.13a | 3.33 ±0.18b | 2.19 ±0.44a |
| soluble sugar content (mg g$^{-1}$) | 30.02 ±3.67a | 44.13 ±3.41b | 14.19 ±1.77c |
| soluble protein content (mg g$^{-1}$) | 6.28 ±1.40a | 12.24 ±0.26b | 7.92 ±0.46a |
| MDA content (μmol g$^{-1}$) | 24.02 ±0.47a | 35.07 ±3.12b | 23.68 ±0.50a |
| gametophyte germination (%) | 82.93 ±10.00a | 100.00 ±0.00a | 98.33 ±2.36a |
| gametophyte increment | 1.54 ±0.18a | 1.82 ±0.40ab | 2.37 ±0.05b |
| gametophyte vigour index | 1.28 ±0.15a | 1.82 ±0.40ab | 2.33 ±0.05b |

Data are average ±1 SE, different letters indicate significant differences ($P < 0.05$) among the three species.

### 3.2 Effect of storage temperature on vegetative propagation of mosses

The three moss species began to germinate at different time. *B. unguiculata* germinated on the eleventh day of inoculation. *D. vinealis* and *D. tectorum* germinated on the sixth day.

The gametophyte germination of all the three mosses after storage changed no more than 20% (Fig. 1a; Table 1). The highest gametophyte germination of *B. unguiculata* was 94.44% at 17 ℃. No significant difference was found between the maximum value and minimum value (75.56%, at 0 ℃). In *D. vinealis*, there was no significantly different gametophyte germination among all storage temperatures, which ranged from 95.56% (0 ℃) to 98.89% (17 ℃). The only significant difference was

between 81.92% and 100% at 0 ℃ and 25 ℃, respectively, in the gametophyte germination of *D. tectorum* after being stored.

The changes of gametophyte increment were all more than 20% after being stored, except a slight decrease of 6.57% in *D. tectorum* at 30 ℃ (Fig. 1b; Table 1). After storage, the most gametophyte increment of *B. unguiculata* was 1.11 at 4 ℃, while the least gametophyte increment was 0.81 at 25 ℃.

Except the significant difference between 4 ℃ and 25 ℃, no significantly different gametophyte increment was found in *B. unguiculata* after being stored. Similarly, no significantly different



gametophyte increment of *D. vinealis* was observed among all storage temperatures. The maximum and minimum of gametophyte increment after storage were 1.03 and 1.23 at 0 ℃ and 17 ℃, respectively. A bigger variation of difference was presented in *D. tectorum* at all storage temperatures except the gametophyte increment between 0 ℃ and 4 ℃. The maximum gametophyte increment of *D.*

*tectorum* was 3.74 at 17 ℃ after being stored, and the minimum value was 1.32 at 0 ℃.

The gametophyte vigour index of the three moss species showed significant changes in a 40-day storage period (Table 2). The largest change of gametophyte vigour index after being stored was displayed in *D. tectorum* with a range from 53.36% decrease (0 ℃) to 57.32% increase (17 ℃). No significant change was found in the gametophyte vigour index of *D. vinealis* among the five

temperatures. However, these gametophyte vigour indexes were all significantly lower than that before storage and decreased by 32.86% (17 ℃) to 45.65% (0 ℃). After being stored, the gametophyte vigour indexes of *B. unguiculata* decreased the least by 18.81% at 4 ℃ and the most by 49.20% at 25 ℃, which indicated the change of *B. unguiculata* was between *D. vinealis* and *D. tectorum*.

After the 40-day storage at the five temperatures, the highest gametophyte germination of *B.*

*unguiculata* and *D. vinealis* were presented at 17 ℃, while the peak in *D. tectorum* was presented at 25 ℃. The highest gametophyte increment of *B. unguiculata* was at 4 ℃ and the peak in *D. vinealis* and *D. tectorum* were both at 17 ℃ as well as the gametophyte vigour index.

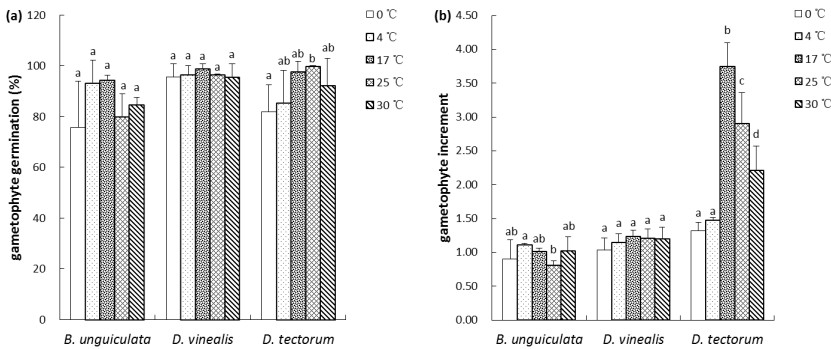

**Fig. 1.** Data (average ± 1 SE) of the three moss species in **(a)** gametophyte germination and **(b)** gametophyte increment after the 40-day storing at five temperatures. Different letters indicate significant differences ($P < 0.05$) among the five temperatures within the same species.

**Table 2** Gametophyte vigour index of the three mosses at different treatments

| treatment | *B. unguiculata* | *D. vinealis* | *D. tectorum* |
|---|---|---|---|
| initial value | 1.28 ±0.15a | 1.82 ±0.40a | 2.33 ±0.05a |
| 0 ℃ | 0.68 ±0.22b | 0.99 ±0.17b | 1.09 ±0.10b |
| 4 ℃ | 1.04 ±0.02ac | 1.11 ±0.13b | 1.26 ±0.03b |
| 17 ℃ | 0.95 ±0.05c | 1.22 ±0.10b | 3.66 ±0.35c |
| 25 ℃ | 0.65 ±0.06b | 1.17 ±0.13b | 2.90 ±0.46a |
| 30 ℃ | 0.86 ±0.18bc | 1.15 ±0.17b | 2.04 ±0.33a |

Data are average ±1 SE, different letters indicate significant differences ($P < 0.05$) among treatments within the same species.




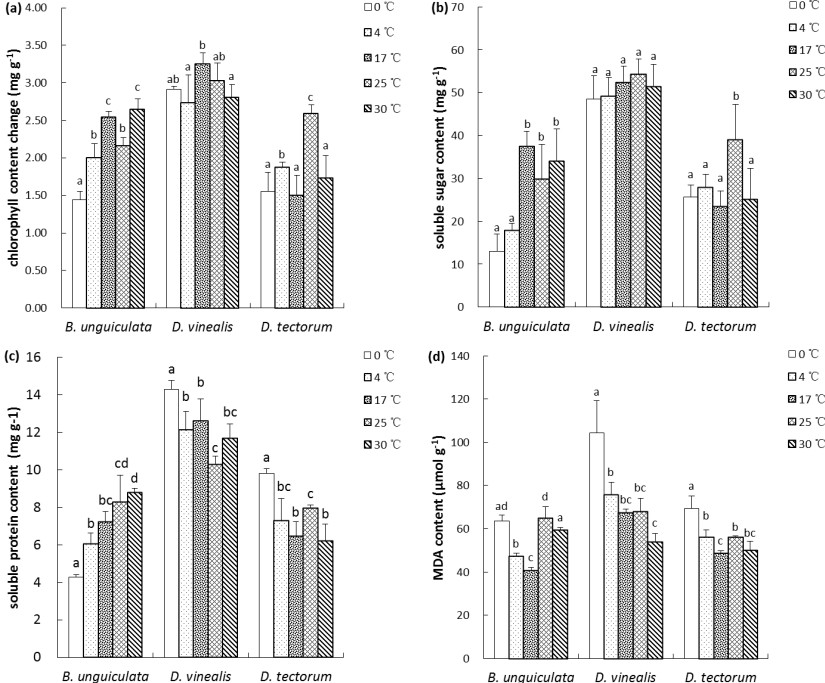

**Fig. 2. (a-d)** Data (average ± 1 SE) of the three moss species in **(a)** chlorophyll content, **(b)** soluble sugar content, **(c)** soluble protein content and **(d)** MDA content after the 40-day storing at five temperatures. Different letters indicate significant differences ($P < 0.05$) among the five temperatures within the same species.

**Table 3** Correlation coefficients among physiological indexes and germination parameters of the mosses in different treatments

| Variables | chlorophyll | sugar | protein | MDA | germination | increment |
|---|---|---|---|---|---|---|
| sugar | .762[**] | | | | | |
| protein | .747[**] | .781[**] | | | | |
| MDA | .220 | .402[**] | .510[**] | | | |
| germination | .473[**] | .414[**] | .313[*] | -.022 | | |
| increment | -.239 | -.187 | -.249 | -.344[*] | .388[**] | |
| vigour index | -.158 | -.122 | -.191 | -.328[*] | .441[**] | .995[**] |

chlorophyll: chlorophyll content; sugar: soluble sugar content; protein: soluble protein content; MDA: MDA content ; germination: gametophyte germination; increment: gametophyte increment; vigour index: gametophyte vigour index.

* indicate significant correlation at $P < 0.05$; ** indicate significant correlation at $P < 0.01$.

5   **Table 4** Grey incidence degree between physiological indexes and gametophyte vigour index of the three mosses in different treatments

| | X1 | X2 | X3 | X4 |
|---|---|---|---|---|
| *B. unguiculata* | 0.60 ±0.20 | 0.57 ±0.20 | 0.57 ±0.22 | 0.77 ±0.20 |
| *D. vinealis* | 0.55 ±0.27 | 0.62 ±0.23 | 0.74 ±0.28 | 0.73 ±0.22 |
| *D. tectorum* | 0.66 ±0.21 | 0.62 ±0.17 | 0.70 ±0.25 | 0.76 ±0.27 |

X1: chlorophyll content; X2: soluble sugar content; X3: soluble protein content; X4: MDA content.





**3.3 Effect of storage temperature on physiological index of mosses**

As shown in Table 1 and Fig. 2a, chlorophyll content of *B. unguiculata* was increased after being stored at four of the five temperatures except 0 ℃. Chlorophyll content of *B. unguiculata* showed an increase trend along with the storage temperature, and a maximum increase of 73.08% was presented at

30 ℃. The smallest change of chlorophyll content was found in *D. vinealis* with the maximal decrease was 17.89% at 4 ℃ and the minimal decreases was 2.39% at 17 ℃. Chlorophyll content of *D. tectorum* after storage was reduced by 31.51% at 17 ℃ and increased by 18.50% at 25 ℃, which were the highest content and lowest content, respectively.

The similar trend to increase as temperature rose also found in the soluble sugar content (Fig. 2b).

The soluble sugar content was higher than any before storage, except the content of *B. unguiculata*, which decreased by 56.52% and 40.47% at 0 ℃ and 4 ℃, respectively (Fig. 2b; Table 1) The soluble sugar content of *D. vinealis* was observed less variation than the other species. No significant difference was found between the maximal and minimal increase, which were 9.92% at 0 ℃ and 23.14% at 25 ℃, respectively. The greatest change of soluble sugar content was presented in *D. tectorum*,

which increased by more than 65% under all storage temperatures.

MDA content had a more significant variance with increasing by more than 50% in all stored gametophytes (Fig. 2d; Table 1). MDA content of *B. unguiculata* and *D. tectorum* both decreased as temperature rise from 0 to 17 ℃ and then increased. Both *B. unguiculata* and *D. tectorum* had a minimal MDA content at 17 ℃. However, the MDA content of *D. vinealis* continuously decreased to

the minimum at 30 ℃.

Some temperature could cause soluble protein content significantly changed (Fig. 2c; Table 1). The soluble protein content of *B. unguiculata* decreased abruptly from the value 40.06% above to the value 31.79% below with the temperature decrease. On the country, soluble protein showed an opposite trend of change in *D. vinealis* and *D. tectorum*. Both of the two species were presented the maximal

increase at 0 ℃, which were 16.64% in *D. vinealis* and 23.65% in *D. tectorum*. The least soluble protein content of *D. vinealis* and *D. tectorum* were reported a decrease by 16.00% at 25 ℃ and a decrease by 21.38% at 30 ℃, respectively.

Our results indicated that the fastest change of chlorophyll content and soluble protein content was in *B. unguiculata* as temperature rose, and the contents of soluble sugar and MDA changed more

rapidly than that of *D. vinealis* and *D. tectorum*, respectively (Fig. 2a-d; Table 1). *D. vinealis* showed slower change of chlorophyll, soluble sugar and soluble protein contents than the other two species. The MDA content, however, varied rapidly with temperature. The biggest change that soluble sugar and MDA increase was in *D. tectorum* after the 40-day storage. Finally, in all three moss species, the maximal index of change was MDA content and the second-most change was in soluble sugar content

(Fig. 2b, 2d; Table 1).

**3.4 Relationship between physiological characteristics and vegetative propagation of mosses**

After analyzing the correlation among physiological indexes and germination parameters of desiccation-tolerant mosses, significant correlation ($P < 0.01$) was found among the four physiological indexes and the three germination parameters, except chlorophyll content and MDA content (Table 3).



The gametophyte germination showed a significant correlation ($P < 0.05$) with soluble protein content and more significant correlation ($P < 0.01$) with chlorophyll content and soluble sugar content. Only MDA content was found a significant negative correlation ($P < 0.05$) with gametophyte increment and gametophyte vigour index.

When the distinguishing coefficient was 0.5, grey incidence degree between physiological indexes (X1: chlorophyll content; X2: soluble sugar content; X3: soluble protein content; X4: MDA content) and gametophyte vigour index of the three moss species were (1) X4 > X1 > X2 = X3 in *B. unguiculata*, (2) X3 > X4 > X2 > X1 in *D. vinealis* and (3) X4 > X3 > X1 > X2 in *D. tectorum* (Table 4).

**4 Discussion**

**4.1 Effect of storage temperature on vegetative propagation of mosses**

For more than a century, researchers have studied many aspects of mosses, such as inocula, pre-treatment (e.g. storage and sterilization), culture methods and culture conditions (Duckett et al., 2004; Hoffman, 1966). Some of these studies implied that physiological characteristics of moss gametophytes were closely related to the success of artificial cultivation, for instance pretreatment with

sucrose and/or abscisic acid could facilitate viability of mosses by increasing DT (Burch and Wilkinson, 2002). In fact, DT is not a constant feature of mosses, like seasonal variation in the desiccation responses (Dilks and Proctor, 1976). In line with previous studies, this study also indicated different results of gametophyte regeneration within same species after desiccation at different temperatures (Fig. 1a, 1b; Table 2), which was probably related to species-specific DT. The gametophyte vigour index of

*D. tectorum* was the most sensitive to change of storage temperature. Contrarily, the gametophyte vigour index of *D. vinealis* was least changed and was not significantly different under storage temperature levels. The vegetative propagation of mosses could be summarily described by the gametophyte vigour index, on the basis of Eqs. (1) - (3) and Table 3. Thus, the effect of storage temperature on vegetative propagation of *D. tectorum* was the biggest in contrast to *D. vinealis*.

Particularly, although the 40-day storage adversely affected regeneration in most moss inocula (Fig. 1a, 1b; Table 1), some inocula of *D. tectorum* stored at 17 ℃ and 25 ℃ produced more new individuals than before. It was not clear whether the enhancement of regeneration was correlated with low-temperature tolerance of *D. tectorum*. In other words, *D. tectorum* possibly suffered low-temperature stress in early winter. Meanwhile, higher temperature (like 30 ℃) also injured inocula

of *D. tectorum*, which implied extreme temperatures were unsuitable for storing moss. It is assumed that further hypothesis could be made about the impact of storage environment to desiccation-tolerant mosses. For example, Burch (2003) found survival and regeneration of dehydration protonemata were reduced after cryopreservation, which was related to damage caused by intra-cellular ice crystal. Desiccation time could also affect restorability of vegetative propagation and physiological

characteristics in desiccation-tolerant mosses (Keever, 1957; Proctor, 2001). In conclusion, some changes caused by environment and/or time occurring in dormant cells could yield different restoration results after rehydration.





### 4.2 Effect of storage temperature on physiological characteristics of mosses

MDA, an important product of membrane lipid peroxidation, increased in all mosses which showed that the 40-day storage caused cell damage (Fig. 2d; Table 1). Hence, soluble sugar content increased correspondingly for protecting membranes and proteins in the dried gametophytes (Fig. 2b; Table 1), as

sugars are mainly substance of stabilizing protein structure below 0.3 (g $H_2O$) (g dry weight)$^{-1}$ in desiccation-tolerant cell (Hoekstra et al., 2001). Conversely, the soluble sugar content of *B. unguiculata* at 0 ℃ and 4 ℃ decreased after being stored. The reason could be that low temperature prevented the conversion from starch to soluble sugar (Pressel et al., 2006). When mosses suffered oxidative damage, the increase of chlorophyll content and soluble protein content in some gametophytes was related to

recovery ability of desiccation-tolerant cell (Fig. 2a, 2c; Table 1). Researchers found that chlorophyll content of mosses increased during desiccation and their photosynthetic capacity recovered rapidly after rewetting (Alpert, 1988; Csintalan et al., 1999) as well as protein synthesis after rehydration (Oliver, 1991), as cellular recovery is an important part of DT (Proctor et al., 2007).

The recovery of *B. unguiculata* on photosynthesis and protein synthesis was facilitated by higher

temperatures (not more than 30 ℃) (Fig. 2a, 2c), which offered an opposite illusion that viability of other mosses tended to be weaker with increased temperature (Hearnshaw and Proctor, 1982). However, the increase trend of MDA content from 17 to 30 ℃ implied that more membrane damage may be caused by storage temperature above 30 ℃ (Fig. 2d). The adverse effects of relatively high temperature in *D. vinealis* and *D. tectorum* were clearly reflected by slower recovery of photosynthesis and protein

synthesis (Fig. 2a, 2c). Although the change of MDA content in *D. vinealis* showed faster repair of cell membrane as the temperatures rose, the moss species possibly had stronger tolerance under the protection of abundant sugars when the recovery of photosynthesis and protein synthesis was slower (Fig. 2a-d).

The response of the three species to temperatures on physiological characteristics reflected

different restorability in a short rehydration time. If rewetting periods were longer than 30 days in the cultivation, the result of vegetative propagation could be defined as a long-term recovery of mosses. Thus, the long-term effect of cell recovery during short-term rehydration could be explained by relationship between physiological characteristics and vegetative propagation of desiccation-tolerant mosses.

### 30 4.3 Relationship between physiological characteristics and vegetative propagation of mosses

Before the storage, the four physiological indexes of gametophytes showed significant difference between *D. vinealis* and *D. tectorum*. However, no significant difference between the two species was observed in regard to the three germination parameters (Table 1). It could be seen that similar fertility between mosses was accompanied by significantly different physiological characteristics. Then

species-specific DT made the vegetative propagation among species present bigger difference than before, which was shown by the gametophyte vigour indexes under the same treatment (Table 1; Table 2). Therefore, the recovery ability of development and regeneration of dried mosses might play a more beneficial role to screen suitable inocula than fresh ones. Although many researches indicated that desiccation-tolerant mosses could recover from drying when they are rehydrated (Csintalan et al., 1999;



Pressel et al., 2006), the overlong desiccation made mosses fail to germinate (Keever, 1957). This study also showed that cell was subjected to oxidative damage after the 40-day desiccation (Fig. 2d; Table 1). At the same time, the regenerative capacity of the three species was declined (Table 2), which implied that membrane integrity and/or other factors had an effect on vegetative propagation of 5 desiccation-tolerant mosses.

Based on the correlation coefficients among physiological indexes and germination parameters of desiccation-tolerant mosses (Table 3), the gametophyte germination revealed significant positive correlation with the chlorophyll content, soluble sugar content and soluble protein content. On the country, the gametophyte increment and gametophyte vigour index were only significant negative 10 correlation with the MDA content. It was a possible reminder that metabolic repair was favorable to germination of new gametophyte and the result of long-term recovery depended more on cell integrity. Therefore, in order to compare the effects of the four physiological indexes on vegetative propagation, the grey incidence degree between physiological indexes and gametophyte vigour index of the three moss species were calculated by Eqs. (4)-(6). As shown in Table 4, the effect of MDA content on 15 gametophyte vigour index was the greatest in *B. unguiculata* and *D. tectorum*, and the incidence degree of MDA in *D. vinealis* was quite similar to the maximum (the former was 0.73 and the latter was 0.74). The MDA content of the three mosses increased as the storage temperature decrease from 17 to 0 ℃, when smaller gametophyte vigour index of *D. vinealis* and *D. tectorum* presented at 0 ℃ and 4 ℃ rather than 25 ℃ and 30 ℃ (Fig. 2d; Table 2). It could be indicated that more membrane damage at 20 low temperature caused the regenerative capacity decline. In addition, the higher gametophyte vigour indexes of *D. tectorum* at 17 ℃ and 25 ℃ than before was possibly related to the reduction of intra-cellular ice crystal during the storage period (Burch, 2003), which facilitated faster recovery upon rehydration than fresh gametophytes (Table 2). However, there were an increasing number of negative influences with increasing temperature presented in the physiological characteristics (Fig. 2a-c). These 25 high temperatures were unfavorable to the recovery of mosses (Hearnshaw and Proctor, 1982). When cell suffered damage under desiccation and temperature stress, the protection of more sugars was particularly important to maintain cell integrity in dry state (Fig. 2d; Table 1). The possible reason for this is that *D. vinealis* showed no significant difference in the regenerative capacity as the cellular protection was equivalent despite different temperatures.

30 Researchers summarized the recovery mechanism of mosses upon rehydration, such as rapid recovery of photosynthesis, respiration and protein synthesis within minutes to hours (Proctor et al., 2007). However, the recovery of carbon balance, cell cycle and the cytoskeleton required more than 24 hours (Alpert and Oechel, 1985; Mansour and Hallet, 1981; Pressel et al., 2006). Based on these results, cell integrity was supposed to be more difficult to recovery than physiological reaction and had a great 35 limit on recovery and regenerative capacity of desiccation-tolerant mosses. During long-term desiccation, cumulative damage affected cell function and integrity over time (Proctor, 2001), which might result in different regenerative capacity of mosses with varied storage time (Keever, 1957). The process was influenced by temperatures that might enhance or suppress cell damage according to the research. According to the above analysis, cell integrity may be a critical influencing factor on the 40 vegetative propagation of mosses.





**5 Conclusions**

The conducted experiment explored the effect of storage temperature on the vegetative propagation of desiccation-tolerant mosses and critical influencing factors. The results indicated that the decline of regenerative capacity in mosses was related to cell damage caused by dehydrated storage. The storage

temperatures during dehydration also influenced vegetative propagation of mosses because of temperature-induced changes in moss cell activity. A further analysis showed the effect of membrane damage on vegetative propagation was the maximal. Meanwhile, soluble sugars increased for protecting cells highlighting the important role cell integrity played in physiological characteristics and vegetative propagation of desiccation-tolerant mosses. In this study, the optimal storage temperature of

*D. vinealis* and *D. tectorum* was 17 ℃, while the suitable temperature was 4 ℃ for *B. unguiculata*. Different responses to the temperatures in the three moss species were linked with species-specific DT, which could guide future research to study some suitable storage methods of inoculation material on the artificial cultivation of moss biocrusts.

     In general, properties of inoculation material are key factors effecting the development and

recovery of moss biocrusts, such as species, physiological feathers and/or other factors. The results helped to partly explain influencing factors on vegetative propagation of desiccation-tolerant mosses and furthermore to offer a new view about fast experimental approach to screen suitable inocula.

*Acknowledgements*. The research was supported by the National Natural Science Foundation of China (grant NOs. 41571268, 41271298). We also express our gratitude to the anonymous reviewers and

editors for their constructive comments and suggestions.

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
