# Peer review of "Effects of storage temperature on the physiological characteristics and vegetative propagation of desiccation-tolerant mosses"

_Biogeosciences, 2017_

## Referee Comment (RC1) · Anonymous Referee #1 · 6 Oct 2017

The authors present a very nice paper looking at the effects of dry storage temperature of three biocrust moss species on viability of the mosses in terms of regeneration and some physiological attributes. They demonstrate that optimal storage temperature varies by species, and can have an impact on the heath and gametophyte reproduction. This is a nice addition to the literature, as mosses contribute great ecosystem benefit to drylands, and can be used in restoration. Dry storage is essential to this endeavor. It also speaks to how much more there is to learn about the ecology of these dessication tolerant species. I highlight below a few issues where more information, clarification or interpretation need to be addressed. A careful edit for proper English grammar could be benefit, but overall the manuscript is well written. Methods Line 23, page

3: "average accumulated temperature...3733 and 3283C" is an odd way to share the temperature range. Instead, please present mean annual temp, mean annual high and mean annual low. How were mosses collected? How were moss storage temperatures maintained? Were they in incubation chambers? I'm unclear on your sampling/splitting design. Did you collect from one colony and split this many ways (for initial, and then the 5 temperature gradients)? You say you have 3 duplicates or subsamples. Does this mean you originally collected from 3 colonies per species, or you split the one colony into "replicates" for each temperature level? Germination parameters: what do you mean by "5 inocula" (line 39 page 4). Does this mean 5 stems?

Results For the physiological parameters, it might be more helpful to say the change from the initial condition, rather than the total. In this way, we can look at positive or negative effects of storage more easily. The grey incidence analysis is over-interpreted. Most of the values overlap and thus, cannot be interpreted as greater or lesser than one another.

---

## Referee Comment (RC2) · Anonymous Referee #2 · 19 Oct 2017

This paper examines the influence of dry storage temperature on regeneration and physiology in three DT mosses from the Loess Plateau. It is a relatively simple study, but does provide important information for moss cultivation in a restoration context, though because of species-specific responses, pre-treatment environmental effects, and RH considerations during drying and dry periods, may not be widely generalizable.

Specific comments follow.

Abstract: L14. I think you mean temperature "levels" L28. cell injury seems vague here. Perhaps mention when you discuss what MDA is above

Introduction In general, you miss out on some key background research by Stark and

Greenwood, who have been examining desiccation and rehydration in Syntrichia for years. L36. "soil fertility accumulation" is an odd phrase P2 L3. "culturing artificially" perhaps should be transposed? L6. What is this "theory"? Is this necessary to say? Paragraph starting with "Desiccation tolerance.." is hard to follow. There seems to be too many ideas in it, and the info on Grimmia seems oddly specific L35. Omit sentence beginning with "Actually.."

Methods Collection: Were they all growing together when they were collected? Were the different species in different microclimates? P3L25. How long did it take moss to dry? Was it different for each species? What was the RH? These are crucial points that relate to regeneration. L37. What was the equilibrating RH during storage? Also, I am unclear on the actual function of the ziploc baggies here. P5L10. Was 25 days the entire length of the regeneration study then? L11. Save for results. L12. Anaogy with seed germination is an interesting idea, but I think you're missing out on key life stages that are missing in angiosperms, like protonema. Was protonemal presence / extent quantified? What about gemmae?

Results Fig.1A is hard to interpret. Are the bars totals after the 25 day regeneration period? Table 1 and Fig 2 kind of go together, and I wished to be able to compare them more easily. Is there a way to incorporate the initial values into Fig. 2 or at least place the table closer to it? Table 4: Why not label the columns with the physiological indexes?

Discussion Careful with over-use of adverbs (Contrarily, Particularly) that don't improve sentences. Overall, while the separate sections are nice, the organization within them is a bit challenging. For example, L35 I don't think a conclusion is appropriate here. Also, in section 4.3 and others I'm noticing less time is spent discussing the current work, and more is spent bringing in related work. It begins to get cumbersome, and the reader loses sight of the key results. A general reframing to focus on key results would be helpful.

Discussion L6-7. I don't understand what the point of this sentence is.

Notes on select specific BG criteria: The paper presents some novel data, but the scope is limited. Much of the scientific methods are valid and outlined well, although the authors miss out on specific drying and storage conditions that could have influenced results more than temperature. Language could be more fluent and precise in numerous places.

---

## Referee Comment (RC3) · Anonymous Referee #3 · 24 Oct 2017

When I read title and abstract of the paper, my first impression was that the mosses were stored in a field-wet state. In became clear just in the M&M section that the mosses were stored air-dried and hermetically sealed. I suggest to mention that point in the abstract. I think it would also be a great idea to get an impression of the relative humidity during storage.

The success of incubation experiments often depends on how well experimental conditions match the niche requirements of the target organisms, in particular those with narrow ecological amplitudes. For example, low gametophyte increment, germination rate and delayed initial germination of Barbula unguiculata does not necessarily mean

that this species generally is outperformed by the Dedymodon species. It may also indicate that the experimental conditions better matched the ecological requirements of the latter, and that other experimental conditions may show a different picture.

Hence, I miss in the paper some discussion of the ecological niche requirements of the particular species investigated. For example, Barbula unguiculata Hedw. and Didymodon vinealis (Brid.) Zander var. vinealis differ with their requirements to light: While both of these species prefer open lands, Barbula unguiculata Hedw. may grow in shadowed areas with down to 30% of relative light intensity, whereas Didymodon vinealis (Brid.) Zander var. vinealis does not develop at relative light intensities below 50% (ISBN-13: 978-3825281045). As the samples were taken at north facing slopes, which possibly receive shadow, I recommend to consult the botanical literature and to consider ecological niche requirements in the discussion of implications for the practice. Further, a more precise description of the sampling procedure and sampling spots might be helpful.

Minor remarks

M&M

p. 4 l. 5 ff.: The weights of 100 and 50 mg of sample for sugar and chlorophyll measurement seem little to ascertain representative sampling. How many replicates were analysed?

p. 5 l. 17.: Please check the correct usage of the terms "seed" and "hypocotyl" in conjunction with mosses. Again, I recommend to consult the botanical literature to be more precise.

Results

Figure 2: I needed to switch between Table 1 and Figure 2 to compare initial values with the temperature effect. I would find Figure 2 easier to comprehend if the initial values could be somehow depicted there (as horizontal lines?).

[Figure]

---

## Author Comment (AC1) · 4 Nov 2017

The authors present a very nice paper looking at the effects of dry storage temperature of three biocrust moss species on viability of the mosses in terms of regeneration and some physiological attributes. They demonstrate that optimal storage temperature varies by species, and can have an impact on the heath and gametophyte reproduction. This is a nice addition to the literature, as mosses contribute great ecosystem benefit to drylands, and can be used in restoration. Dry storage is essential to this endeavor. It also speaks to how much more there is to learn about the ecology of these dessication tolerant species. I highlight below a few issues where more information, clarification or interpretation need to be addressed. A careful edit for proper English grammar could be benefit, but overall the manuscript is well written.

Thank you for your highly praise and helpful comments on our manuscript. We will modify English in the revised manuscript and address each issue below:

Methods Line 23, page 3: "average accumulated temperature...3733 and 3283C" is an odd way to share the temperature range. Instead, please present mean annual temp, mean annual high and mean annual low.

Agreed. We will add the three temperatures in the revised manuscript.

How were mosses collected? How were moss storage temperatures maintained? Were they in incubation chambers? I'm unclear on your sampling/splitting design. Did you collect from one colony and split this many ways (for initial, and then the 5 temperature gradients)? You say you have 3 duplicates or subsamples. Does this mean you originally collected from 3 colonies per species, or you split the one colony into "replicates" for each temperature level?

In fact, we stored samples at 0 ° C and 4 ° C in two refrigerators, respectively. Other samples were stored at 17 ° C, 25 ° C and 30 ° C in three growth chambers, respectively. In our sampling design, we collected moss crusts of a given species from many colonies, and then moss crusts were packed in 3 ziplock baggies. After stored, we collected some gametophytes from moss crusts as subsamples. Thus, we will revise the sentence.

Germination parameters: what do you mean by "5 inocula" (line 39 page 4). Does this mean 5 stems?

Yes. The "5 inocula" means five 2-mm stems of living mature gametophytes.

Results For the physiological parameters, it might be more helpful to say the change from the initial condition, rather than the total. In this way, we can look at positive or negative effects of storage more easily.

We also consider that readers can look at effects of storage from changes of physiological parameters more easily. However, we found bigger standard errors than true value appearing in figures of change percentages. It might cause misunderstanding. Therefore we described percentages of parameter changes in words. In addition, Reviewer #3 suggested adding the initial values (depicted by horizontal lines) into Fig. 2, which may be easier to comprehend.

The grey incidence analysis is over-interpreted. Most of the values overlap and thus, cannot be interpreted as greater or lesser than one another.

We tried to quantify the impact of physiological parameters on vegetative propagation by

using grey incidence analysis, which can help to determine if there were different impacts between physiological parameters. We think the method is more suitable the analysis of moss vegetative propagation with few information. Bigger incidence degree meant relatively more impact on vegetative propagation, and even little difference might be meaningful in a grey system. Nevertheless, more precise method or model will be required to quantify the impact of physiological parameters in further studies.

---

## Author Comment (AC2) · 4 Nov 2017

This paper examines the influence of dry storage temperature on regeneration and physiology in three DT mosses from the Loess Plateau. It is a relatively simple study, but does provide important information for moss cultivation in a restoration context, though because of species-specific responses, pre-treatment environmental effects, and RH considerations during drying and dry periods, may not be widely generalizable.

Thank you for carefully reading and many helpful comments. We will consider and response every comments below:

Abstract: L14. I think you mean temperature "levels"

Yes. It will be changed in the revised manuscript.

L28. cell injury seems vague here. Perhaps mention when you discuss what MDA is above

The decrease of soluble sugar may cause cellular protein denaturation upon desiccation. Thus, the phrase "cell injury" actually included not only membrane damage showed by MDA, but also protein injury showed by soluble sugar.

Introduction In general, you miss out on some key background research by Stark and Greenwood, who have been examining desiccation and rehydration in Syntrichia for years.

Thank you for providing important information about DT. They will be added to the revised manuscript.

L36. "soil fertility accumulation" is an odd phrase

Agreed. It will be revised.

P2 L3. "culturing artificially" perhaps should be transposed?

Agreed. It will be revised.

L6. What is this "theory"? Is this necessary to say?

No. There is little theory research on vegetative propagation of mosses compare with sexual reproduction. We will delete the word.

Paragraph starting with "Desiccation tolerance.." is hard to follow. There seems to be too many ideas in it, and the info on Grimmia seems oddly specific

Agreed. The information about DT should be simplified. In recently years, some researchers (e.g. Stark et al. 2005) studied DT by culturing shoots, which guided us to consider impact of DT on vegetative propagation. However, we have some logistic problems and they will be revised.

L35. Omit sentence beginning with "Actually.."

Agreed. It will be revised.

Methods Collection: Were they all growing together when they were collected? Were the different species in different microclimates?

The three species were collected from different plots. Unfortunately, we do not have any data

about the microclimates, though we collected a given species in same plot.

P3L25. How long did it take moss to dry? Was it different for each species? What was the RH? These are crucial points that relate to regeneration.
We dried all three species for 24-48 hours. Unfortunately, we do not have the data on RH during drying. Nevertheless, most of gametophytes were dry (e.g. Figure 1) when we collected moss crust. We believed there was little effect caused by RH during drying.

[Figure]

Figure 1 *D. vinealis* before collected

L37. What was the equilibrating RH during storage? Also, I am unclear on the actual function of the ziploc baggies here.
The equilibrating RH was 55% and will be add to the revised manuscript. On account of we stored mosses in refrigerators or growth chambers (the detail can also read in Answer to Reviewer #1) with different RH, the ziplock baggies were used for preventing water from air.

P5L10. Was 25 days the entire length of the regeneration study then?
After new gametophytes germinated, we continued culturing for 25 days. Thus, the entire lengths of the regeneration study were 30 days in *D. vinealis* and *D. tectorum*, and 35 days in *B. unguiculata*.

L11. Save for results.
It will be revised.

L12. Anaogy with seed germination is an interesting idea, but I think you're missing out on key life stages that are missing in angiosperms, like protonema. Was protonemal presence / extent quantified? What about gemmae?
In fact, we ever tried to measure the timing of protonemal production and protonemal growth rate in trial tests. Nevertheless, mosses protonema germinated lately made it difficult to

differentiate from soil. Furthermore, there were not gemmae in three species except *Didymodon tectorum*.

Results Fig.1A is hard to interpret. Are the bars totals after the 25 day regeneration period? Fig. 1 shows results of fifth observation. Thus, the bars are totals after full regeneration period. We will revise description of the figure.

Table 1 and Fig 2 kind of go together, and I wished to be able to compare them more easily. Is there a way to incorporate the initial values into Fig. 2 or at least place the table closer to it? Reviewer #3 suggested adding the initial values (depicted by horizontal lines) into Fig. 2, which may be easier to comprehend.

Table 4: Why not label the columns with the physiological indexes? Agreed. It will be revised.

Discussion Careful with over-use of adverbs (Contrarily, Particularly) that don't improve sentences. Overall, while the separate sections are nice, the organization within them is a bit challenging. For example, L35 I don't think a conclusion is appropriate here. Also, in section 4.3 and others I'm noticing less time is spent discussing the current work, and more is spent bringing in related work. It begins to get cumbersome, and the reader loses sight of the key results. A general reframing to focus on key results would be helpful. Thank you for your comments in language and organization. We will make effort to improve English and revise the organization in the revised manuscript.

Discussion L6-7. I don't understand what the point of this sentence is. After read again, we find this sentence should be deleted.

Notes on select specific BG criteria: The paper presents some novel data, but the scope is limited. Much of the scientific methods are valid and outlined well, although the authors miss out on specific drying and storage conditions that could have influenced results more than temperature. Language could be more fluent and precise in numerous places. Thank you for pointing out mistakes and providing many advices! We will revise the manuscript as your suggestion.

---

## Author Comment (AC3) · 4 Nov 2017

Thank you for helpful comments and constructive suggestions on the manuscript. We will consider and response every comments below:

When I read title and abstract of the paper, my first impression was that the mosses were stored in a field-wet state. In became clear just in the M&M section that the mosses were stored air-dried and hermetically sealed. I suggest to mention that point in the abstract. Agreed. We will add the storage state in the abstract.

I think it would also be a great idea to get an impression of the relative humidity during storage.

We agree with you and Reviewer #2. It will be revised.

The success of incubation experiments often depends on how well experimental conditions match the niche requirements of the target organisms, in particular those with narrow ecological amplitudes. For example, low gametophyte increment, germination rate and delayed initial germination of Barbula unguiculata does not necessarily mean that this species generally is outperformed by the Dedymodon species. It may also indicate that the experimental conditions better matched the ecological requirements of the latter, and that other experimental conditions may show a different picture. Hence, I miss in the paper some discussion of the ecological niche requirements of the particular species investigated. For example, Barbula unguiculata Hedw. and Didymodon vinealis (Brid.) Zander var. vinealis differ with their requirements to light: While both of these species prefer open lands, Barbula unguiculata Hedw. may grow in shadowed areas with down to 30% of relative light intensity, whereas Didymodon vinealis (Brid.) Zander var. vinealis does not develop at relative light intensities below 50% (ISBN-13: 978-3825281045). As the samples were taken at north facing slopes, which possibly receive shadow, I recommend to consult the botanical literature and to consider ecological niche requirements in the discussion of implications for the practice. Further, a more precise description of the sampling procedure and sampling spots might be helpful.

Thank you for insightful comment on ecological niche requirements of mosses. We will add a list of moss species including ecological information and more precise sampling design to the revised manuscript. We believe that different niche requirement (e.g. species-specific DT) will influence the choice of moss inocula on artificial cultivation and biocrust restoration, thus three species were compared in the paper. However, it seems be unclear to readers. We will revise the manuscript.

**Minor remarks**

**M&M**

p. 4 l. 5 ff.: The weights of 100 and 50 mg of sample for sugar and chlorophyll measurement seem little to ascertain representative sampling. How many replicates were analysed? Three replicates were analysed and had 100 mg in every replicate for sugar measurement. Similarly, there was 50 mg in a replicate for chlorophyll measurement.

p. 5 l. 17.: Please check the correct usage of the terms "seed" and "hypocotyl" in conjunction

with mosses. Again, I recommend to consult the botanical literature to be more precise. It will be revised.

Results

Figure 2: I needed to switch between Table 1 and Figure 2 to compare initial values with the temperature effect. I would find Figure 2 easier to comprehend if the initial values could be somehow depicted there (as horizontal lines?). Good idea! We will revise figures as you said.

---

## Author Response (AR2)

[revised manuscript text omitted]

- 5 average monthly precipitation was 11.98 mm, and the average monthly temperature was 9.88 °C (high) to -3.64 °C (low) (Chinese Central Meteorological Station, 2017). Cyanobacteria and mosses dominated the biocrust communities in this region, and the coverage of moss-dominated biocrusts might evencan reach around approximately 80% on north-facing slopes in the study region (Zhao et al., 2014).
- 10

The moss taxa used in the study were Barbula unguiculata, Didymodon vinealis and Didymodon tectorum, which dominated the moss crusts in different the plots. Lots of B. unguiculata dominated in woodland areas and waswere found in shadowed areas and under vegetation coverage, which dominated in the woodland. D. vinealis was widely distributed in the study site under among different water and light environments, - Samples of and the species were collected from croplands that had been

15 abandoned croplands for more than ten years. The dominated dominant vegetation of the plot croplands was grasses; thus, most of moss crusts D. vinealis were exposed to sunlight in the winter. D. tectorum grew on side slopes and sometimes were occasionally collected from under the shade of vascular plants.

**2.2 Experimental design**

- 20 EachSome of the three moss crusts were used to measure initial values of physiological indices (chlorophyll content, soluble sugar content, soluble protein content and MDA content) and germination parameters (gametophyte germination, gametophyte increment and gametophyte vigor index) was separated into two parts as soon as they were immediately following their transported to the laboratory. One part was used to measure initial physiological indices (chlorophyll content, soluble sugar content,
- 25 soluble protein content and MDA content) and germination parameters (gametophyte germination, gametophyte increment and gametophyte vigor index). The other was rest of moss crusts were stored at one of five temperature levels, i.e., 0 °C, 4 °C, 17 °C, 25 °C and 30 °C. All the Each temperatures were was controlled within  $\pm 1$  °C around the target. On the 41st day of storage, Then, the mosses-moss crusts were taken out on the 41st day of storage removed, and the physiological indices and germination 30 parameters mentioned described above were measured.

35

**2.3 Moss crusts storage and mosses collection and storage**

The crusts of tThree species of mosses erusts were collected from many colonies and then air-dried in the shade for 24-48 hours-after being collected from many colonies, although; most of mosses-crust samples were dried in the field. Then, the samples were transported to the laboratory of the State Key Laboratory of Soil Erosion and Dry-land Farming on the Loess Plateau which is-in Yangling, Shaanxi

Province. Samples were stored in one of two refrigerators (at 0  $\,^{\circ}$ C and or 4  $\,^{\circ}$ C) and or one of three growth chambers (at 17 °C, 25 °C and 30 °C). Before storage, Thus, thethe moss crusts were packed had been placed in Ziploc baggiesre-sealable plastic bags to prevent changes in the water content before being stored, and then they were kept. The samples were stored in the dark under a light-blocking fabric. The measurement of the water contents measurements of the moss gametophytes were all less than 10%, and the equilibrating relative humidity during storage was 55% during storage. After the 40-day dry period, some subsamples of desiccated gametophytes were collected as subsamples to measure the physiological indices and germination parameters.

**2.4 Measurement of the physiological indicesex and germination parameters**

**2.4.1 Physiological indicesex**

[revised manuscript text omitted]

gametophyte vigor index = gametophyte germination
$$\times$$
 gametophyte increment (3)

According to Eq. (1) - (3), the gametophyte vigor index could summarily describesummarizes the vegetative propagation of the mosses.

**2.5 Statistical analyses**

5

The differences in physiological indices and germination parameters among treatments and mosses were tested using a-one-way analysis of variance (ANOVA) with Fisher's least significant difference post hoc test (LSD) at P < 0.05. The relationships between the physiological indices and germination parameters of the three moss species were quantified by calculating the Pearson correlation coefficients. These statistical analyses were completed using SPSS 22.0.

The effects of physiological characteristics on vegetative propagation was-were analyzed by a gray incidence analysis in Microsoft Excel 2010 (Deng, 1984; Lin et al., 2009). The gray incidence degree between each of the reference sequences (physiological indices) and the compared sequence (gametophyte vigor index) was calculated by Eq. (4) - (6):

10
$$\Delta_i(k) = |y(k) - x_i(k)|, k = 1, 2, ..., n; i = 1, 2, 3, 4$$
 (4)

$$\xi_i(X_i, Y) = \frac{\min_i \min_k \Delta_i(k) + \rho \max_i \max_k \Delta_i(k)}{\Delta_i(k) + \rho \max_i \max_k \Delta_i(k)}, k = 1, 2, \dots, n; i = 1, 2, 3, 4$$
(5)

$$r_i = \frac{1}{n} \sum_{k=1}^{n} \xi_i(k), k = 1, 2, \dots, n; \ i = 1, 2, 3, 4$$
(6)

where  $\Delta_i(k)$  and  $\xi_i(X_i, Y)$  are the absolute difference and the gray relational coefficient, respectively, between  $X_i$  (physiological indicesindex *i*) and *Y* (gametophyte vigor index) at point *k*. The gray relational coefficient ( $r_i$ ) is between the  $i_{th}$  physiological index and its gametophyte vigor index when

the distinguishing coefficient ( $\rho$ ) is 0.5.

The gray incidence degree is thea sum of the gray relational coefficients.

**3 Results**

5

15

**3.1 The initial state measurement values of the mosses**

20 Table 1 The iInitial values of physiological indicesex and germination parameters in the three mosses

| Index                                         | B. unguiculata             | D. vinealis                | D. tectorum                |
|-----------------------------------------------|----------------------------|----------------------------|----------------------------|
| chlorophyll content (mg g -1 )     | 1.53 ±0.13 a               | $3.33\pm 0.18~\text{b}$    | 2.19 ±0.44 a               |
| soluble sugar content (mg g -1 )   | $30.02 \pm 3.67 \text{ a}$ | 44.13 ±3.41 b              | 14.19 ±1.77 c              |
| soluble protein content (mg g -1 ) | $6.28 \pm 1.40 \text{ a}$  | $12.24 \pm 0.26 \text{ b}$ | $7.92 \pm 0.46 \text{ a}$  |
| MDA content (µmol g -1 )           | $24.02 \pm 0.47$ a         | $35.07 \pm 3.12 \text{ b}$ | $23.68 \pm 0.50 \text{ a}$ |
| gametophyte germination (%)                   | $82.93 \pm 10.00 a$        | $100.00 \pm 0.00 a$        | 98.33 ±2.36 a              |
| gametophyte increment                         | 1.54 ±0.18 a               | $1.82 \pm 0.40 \text{ ab}$ | $2.37 \pm 0.05 \text{ b}$  |
| gametophyte vigor index                       | $1.28 \pm 0.15 a$          | $1.82\ \pm 0.40$ ab        | $2.33\ \pm 0.05\ b$        |
|                                               |                            | . D                        |                            |

Data are average  $\pm 1$  SE, and different letters indicate significant differences (P < 0.05) among the three species.

The three moss species began to germinate new gametophytes from the original inocula at different times, while\_whereas no gametophyte germinated in the control groups in\_as of the last\_final measurement (fifth) (fifth-observation). *B. unguiculata* germinated on the eleventh day of inoculation, so that and the entire length of its cultivation time period was 35 days. *D. vinealis* and *D. tectorum* each germinated on the sixth day, with a 30-day cultivation period. The initial values of the physiological indices and germination parameters of the three mosses was\_are shown in Table 1. It can be seen that the four physiological indices and gametophyte germination of *D. vinealis* were significantly higher

30 than those of the other two species. The biggest largest values of gametophyte increment and gametophyte vigor index were found in *D. tectorum*, and the smallest lowest germination parameters

values were found in *B. unguiculata*. However, no significant differences in the contents of chlorophyll, soluble protein and MDA between *D. tectorum* and *B. unguiculata* were found.

**3.2 Effect of storage temperature on the vegetative propagation of mosses**

The germination times of each of the three mosses and controls after storage at each temperature did

not differ significantly fromwere the same as the initial statevalues, whereas controls still had no gametophyte. In-At the fifth observation, the gametophyte germination of all each of the three species had changed from the initial value by no more than 20% (Fig. 1a; Table 1). The highest gametophyte germination of *B. unguiculata* was 94.44% at 17 °C. No significant difference was found between the maximum value and minimum value (75.56%, at 0 °C). In *D. vinealis*, gametophyte germination did notthere was no significantly different gametophyte germination among all the storage temperatures;

- which ranged \_ and ranged from 95.56% (0 °C) to 98.89% (17 °C). The only significant difference in gametophyte germination was observed in *D. tectorum* and was between 81.92% and 100% after storage at 0 °C and 25 °C, respectively, in the gametophyte germination of *D. tectorum* after being stored.
- 15 The changes of in gametophyte increment were all more than 20% after being stored, storage except in *D. tectorum* at 30 °C, for which for a slight decrease of 6.57% was observed in *D. tectorum* at 30 °C (Fig. 1b; Table 1). After storage, the largest gametophyte increment of *B. unguiculata* was 1.11 at 4 °C, while whereas the smallest gametophyte increment was 0.81 at 25 °C. Except for the a significant difference between 4 °C and 25 °C, no significantly difference in gametophyte increment was observed among the storage temperatures in *B. unguiculata* after being stored. Similarly, no significantly difference in the gametophyte increment of *D. vinealis* was observed among all the storage temperatures. The maximum and minimum gametophyte increments after storage were 1.03 and 1.23 at 0 °C and 17 °C, respectively, *D. vinealis*. A bigger variation of Larger differences in *D. tectorum* at more the storage temperatures was presented were observed in *D. tectorum*.
- all storage temperatures, except for the difference in gametophyte increment between 0 ℃ and 4 ℃.
   The maximum gametophyte increment of *D. tectorum* was 3.74 at 17 ℃ after being storedstorage, and the minimum value was 1.32 at 0 ℃.
  - The gametophyte vigor index of the three moss species showed significant changes in a over the 40-day storage period (Table 2). The largest changes of in gametophyte vigor index after being stored storage was displayed were observed in *D. tectorum*, with the index a rangeranging from a 53.36% decrease (0 °C) from the initial value to a 57.32% increase (17 °C). No significant change difference was found in the gametophyte vigor index of *D. vinealis* among the five temperatures was observed in *D. vinealis*. However, these the index values gametophyte vigor indices were all significantly lower than that the initial value (before storage), representing and decreased by decreases of 32.86% (17 °C)
- 35

30

to 45.65% (0 °C). After being stored storage, the gametophyte vigor index values of *B. unguiculata* decreased the least by 18.81% at 4 °C and the most by 49.20% at 25 °C, which indicated the change of *B. unguiculata* representing changes intermediate was between those of *D. vinealis* and *D. tectorum*.

After the 40-day storage at the five temperatures, the highest gametophyte germination percentages of *B. unguiculata* and *D. vinealis* were at 17 °C, while whereas the peak-highest percentage

in *D. tectorum* was at 25 °C. The highest gametophyte increment of *B. unguiculata* was at 4 °C<del>, and the</del>peak-The highest gametophyte increment values in *D. vinealis* and *D. tectorum* were both at 17 °C, which was the same for as observed for the gametophyte vigor index values of these two species.

**Fig. 1.** Data (average  $\pm 1$  SE) of the three moss species in-on(a) gametophyte germination and (b) gametophyte increment after the 40-day storage period at each of the five temperatures. Different letters indicate significant differences (P < 0.05) among the five temperatures within the same species. Dotted lines represent the approximate values of the three species in-two germination parameters before storage for each species (the true values are shown in Table 1).